# Fragmentation by major dams and implications for the future viability of platypus populations

Jose L. Mijangos [1,2✉], Gilad Bino[3], Tahneal Hawke[3], Stephen H. Kolomyjec [4], Richard T. Kingsford[3], Harvinder Sidhu[1], Tom Grant[3], Jenna Day[5], Kimberly N. Dias[5], Jaime Gongora [5] & William B. Sherwin [6]

The evolutionarily unique platypus (*Ornithorhynchus anatinus*) has experienced major declines and extinctions from a range of historical and recent interacting human-mediated threats. Although spending most of their time in the water, platypuses can move over land. Nevertheless, uncertainties remain whether dams are barriers to movement, thus limiting gene flow and dispersal, essential to evolution and ecology. Here we examined disruption of gene flow between platypus groups below and above five major dams, matched to four adjacent rivers without major dams. Genetic differentiation ($F_{ST}$) across dams was 4- to 20-fold higher than along similar stretches of adjacent undammed rivers; $F_{ST}$ across dams was similar to differentiation between adjacent river systems. This indicates that major dams represent major barriers for platypus movements. Furthermore, $F_{ST}$ between groups was correlated with the year in which the dam was built, increasing by 0.011 every generation, reflecting the effects of these barriers on platypus genetics. This study provides evidence of gene flow restriction, which jeopardises the long-term viability of platypus populations when groups are fragmented by major dams. Mitigation strategies, such as building of by-pass structures and translocation between upstream and downstream of the dam, should be considered in conservation and management planning.

[1] School of Science, UNSW, Canberra, Australia. [2] Centre for Conservation Ecology and Genomics, Institute for Applied Ecology, University of Canberra, Canberra, Australia. [3] Centre for Ecosystem Science, School of Biological, Earth and Environmental Sciences, UNSW, Sydney, Australia. [4] College of Science and the Environment, Lake Superior State University, Sault Sainte Marie, USA. [5] Sydney School of Veterinary Science, Faculty of Science, The University of Sydney, Sydney, Australia. [6] Evolution & Ecology Research Centre, UNSW, Sydney, Australia. ✉email: luis.mijangos@gmail.com

The semi-aquatic platypus (*Ornithorhynchus anatinus*), along with echidnas, belong to the order Monotremata, the most species-poor (*n* = 5) and most basal branch of mammals, which diverged from marsupials and eutherians 187 Mya[1]. Platypuses have a unique combination of features, including oviparity, venomous spurs in males, electroreception used to locate freshwater macroinvertebrates, biofluorescent pelage, and multiple sex chromosomes (five pairs instead of one[2–4]). The uniqueness and rarity of platypus features (*sensu* Pavoine et al.[5]) and its evolutionary distinctiveness[6] make it arguably one of the most irreplaceable mammals existing today.

The platypus is currently listed as 'Near Threatened' by the International Union for Conservation of Nature (IUCN[7]), 'Endangered' in South Australia (*National Parks and Wildlife Act 1972*) and 'Vulnerable' in Victoria[8].

There is increasing evidence of larger numbers of platypuses in historical times[9] and ongoing declines and extinctions of local populations[2,10,11]. Declines have been driven by multiple and synergistic threats, including river regulation, loss and modification of habitats, climate change, pollution, by-catch mortality and predation by invasive species[2,9–11]. Continued declines due to current and future climate change are predicted as a result of increased frequency and severity of droughts[2,12,13], as well as elevated temperature conditions which could lead to the loss of more than 30% of suitable habitat by 2070[12,14].

Threats to freshwater ecosystems are commonly synergistic and are intensified by the construction of major dams that can have immediate and long-term impacts[15]. Nearly half of the world's river discharge is impacted by flow regulation and fragmentation[16]. Dams pose a major threat to global freshwater biodiversity[17]. Large dams form major barriers for aquatic organisms, limiting critical ecological processes, such as fish migration[18]. Water impounds behind major dams form wind-exposed, deep, and standing (lentic) ecosystems which can offer little resources for flow-dependant species[19]. In Australia, dams are one of the more serious threats for platypus conservation, given their potential broad impact on habitat[2,12,20]. Major dams are widespread across much of the platypus' distribution, where as many as 77% (383 out of 495) of the Australian major dams (wall height >10 m; ancold.org.au) coincide within the regions where platypuses occur (Fig. 1a; see also Bino et al.[11]). Immediate adverse effects of major dams extend over large areas both upstream and downstream. Below major dams, altered natural flow regimes, including changing of the timing of flows and important reduction in flow volumes have been found to significantly impact platypus abundances and demographics[21]. Conditions below and above major dams represent poor foraging and burrowing habitat for platypuses, given lower productivity of macroinvertebrate prey species[10,22–25].

Long-term effects of major dams may include reduction in the ability of platypuses to move between potential habitat areas. This fragmentation has twofold impact; first, it restricts the ability to recolonise available habitat or migrate to areas with more suitable conditions[26]. Secondly, fragmentation also simultaneously reduces both local population size and gene flow, each of which is expected to lead to increased inbreeding and reduction of the genetic variation necessary for adaptation to changes including threats[27]. One adverse consequence of small population size is lower survival and lower reproduction output due either to inbreeding depression or to catastrophic stochastic events. Another adverse consequence is reduced variation between individuals, necessary for adaptation to changes such as the threats listed above[28]. These genetic changes may be prevented by immigration because gene flow replenishes the gene pool of populations, but of course, this will only happen if the small population is not a fragmented isolate[29,30].

For platypuses, major dams are predicted to be a barrier for dispersal[31,32], with potential long-term ramifications for gene flow, genetic variation, and adaptation to threats. However, both the restriction of dispersal and the genetic consequences remain largely unquantified. When major dams are assumed to pose barriers for movements, population viability analyses demonstrate considerable impacts by major dams, particularly in synergy with lower habitat quality and droughts, which are projected to increase[11]. In addition, since the introduction of red foxes (*Vulpes vulpes*) to Australia, overland movements of platypuses carry an increased risk of predation[24], effectively increasing the impact of dams as barriers to platypuses. However, the extent to which major dams restrict platypus dispersal remains unclear because landscape connectivity varies due to both the species' life history and landscape features[26]. Platypuses are known to be able to climb around dams up to 10 m high (Dr Tom Grant & Dr Anne Musser, personal communication, June 23, 2021), although their ability to find their way around higher structures is currently unknown. Their ability to swim across the large deep-water impounds above the dam is also unclear.

Therefore, our research uses genetic methods to focus on the connectivity of platypus groups above and below major dams. Genetic-based methods used to infer patterns of dispersal and gene flow[33] commonly examine the positive relationship between the amount of genetic differentiation between populations or individuals and the geographic distance separating them[34]. The presence of a dispersal barrier could be inferred by testing whether populations or individuals, separated by potential barriers, are more genetically differentiated than populations or individuals in landscapes lacking such barriers but separated by a similar distance. Genetic differentiation can increase due to dispersal barriers within one to 15 generations during computer simulations[35], but is unlikely to arise if population size is large (>50 individuals[36]).

To determine whether major dams have reduced dispersal and gene flow between platypus groups, we analysed genetic data from platypuses sampled in nine rivers; five rivers were regulated by major dams, and four were unregulated (Fig. 1). If major dams adversely affected gene flow between platypus groups, we predicted the following: (a) individuals and groups separated by a major dam in a river should be more differentiated than in an unregulated river, and; (b) genetic differentiation across major dams should correlate with the time since the dam was built.

## Results

**Genetic variation within groups**. Mean single nucleotide variation (SNP) genetic variation across all rivers (expected heterozygosity) was $He = 0.140$. $He$ was significantly different between all groups within one river system (except for Severn above the dam/Severn below the dam; *p*-value >0.05; Table 1). $He$ was also significantly different between regions (except for Snowy Rivers/ Upper Murray Rivers; *p*-value >0.05; Table 1). Border Rivers, located in the north, had the lowest $He$ (range: 0.130–0.135), followed by the Snowy Rivers (0.139–0.144) and the Upper Murray Rivers (0.140–0.152), river regions in the south (Fig. 1). Estimates of allelic richness follow the same trend as heterozygosity estimates. Inbreeding estimates ($F_{IS}$) were close to zero except for the microsatellite dataset (Table 1).

**Connectivity between platypus groups—effects of major dams**. For unregulated and regulated river comparisons, the river with the dam showed higher genetic differentiation: Mitta-Mitta above versus below dam had $F_{ST} = 0.024$, whereas Ovens above versus below had $F_{ST} = 0.002$; Nepean below versus above dam had $F_{ST} = 0.073$, whereas Wingecarribee above versus below had

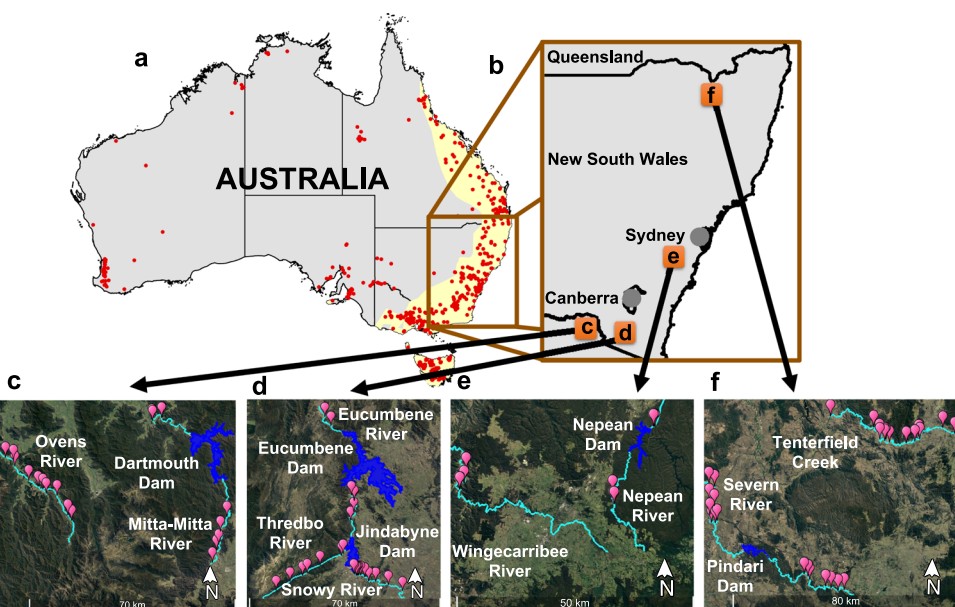

**Fig. 1 Sampling sites. a** Distribution of major dams (>10 m high; red points) within and outside the IUCN platypus distribution (yellow shade), and the focus regions for this study (brown inset). **b** Location of rivers in south-east Australia where platypuses were sampled (orange squares) in rivers that were regulated (with major dams) and unregulated (no major dams). **c** Upper Murray Rivers: Ovens (unregulated) and Mitta-Mitta Rivers (regulated, upstream sections are in the south, confluence with Ovens is out of the frame, in northwest). **d** Snowy Rivers (do not follow the paired experimental design, due to geographic constraints; see methods): Eucumbene (regulated), Thredbo (unregulated), and Snowy River (regulated, Snowy flows downstream to the southeast). **e** Central NSW Rivers: Wingecarribee River (unregulated) and Nepean River (regulated, downstream sections are in the north, there is no confluence with Wingecarribee). **f** Border Rivers: Tenterfield Creek (unregulated) and Severn River (regulated, upstream sections are to the east, confluence with Tenterfield is out of the frame, in northwest). Pink balloons represent the 81 sampling sites; rivers are coloured in light blue, and reservoirs behind major dams are in dark blue. Platypus distribution shapefile was downloaded from The IUCN Red List of Threatened Species[74]. Rivers and dams shapefiles were downloaded from Geoscience Australia[75,76]. Map of Australia shapefile was downloaded from Australian Bureau of Statistics[77]. Dams height and GPS coordinates were downloaded from Australian National Committee on Large Dams Incorporated[78].

$F_{ST} = 0.016$; and Severn below versus above dam had $F_{ST} = 0.061$, whereas Tenterfield above versus below had $F_{ST} = 0.007$ (Table 2). In each case, the dammed versus undammed $F_{ST}$ values differed by more than two standard errors of the mean; the average $F_{ST}$ for the three dammed rivers (0.053) was about six times higher than the paired undammed rivers (0.008). The relatively high within-locality variation for microsatellites has the potential to lower $F_{ST}$ for microsatellites relative to SNPs[37], however, such a trend was not evident—in fact, the opposite trend was seen. Finally, in the more complex Snowy Rivers system (Fig. 1), this simple paired $F_{ST}$ analysis was not easy to interpret, so we relied upon the other analyses presented below. Using Mutual information and Jost's D to assess genetic differentiation with and without major dams gave results that were comparable to those from $F_{ST}$ (Supplementary Tables 3–4 and Supplementary Figure 2).

Over all four river systems, we observed a positive and significant relationship ($R^2 = 0.615$; $p$-value = 0.013) between $F_{ST}$ and the number of platypus generations since the building of the dam (Fig. 2). We note again that potential bias towards lower $F_{ST}$ values in microsatellites than in SNPs, mentioned above, was not evident—the oldest dam was in the river system analysed by microsatellites, and this system showed the highest $F_{ST}$ (Fig. 2).

Spatial differentiation summarised by principal components analysis (PCA) of the Upper Murray Rivers (Mitta-Mitta and Ovens Rivers) did not show complete separation of samples for different locations, but there was noticeable clustering of platypuses into three groups: Ovens river (unregulated); below the dam in the Mitta-Mitta River, and above the dam in the Mitta-Mitta River (Fig. 3a). Snowy Rivers (Snowy, Thredbo and Eucumbene Rivers) did not follow the paired experimental design

due to geographic constraints. PCA analyses showed that platypuses from the Snowy River formed a separated cluster to that of the Thredbo and Eucumbene Rivers (Fig. 3b), whereas platypuses from the two latter rivers overlapped somewhat on the PCA plot. Notably, platypuses from the Eucumbene River above the dam were closer to platypuses from Thredbo River than platypuses from the Eucumbene River below the dam. PCA analyses of the central New South Wales Rivers (Nepean and Wingecarribee Rivers) did not show a clear clustering pattern (Fig. 3c) possibly due to the low number of markers used in this analysis (12 microsatellites) compared to the other rivers systems (2641 SNPs). For the Border Rivers (Tenterfield Creek and Severn River), the principal component analysis (PCA) of these rivers indicated three well-separated clusters (Fig. 3d), with platypuses collected below and above the dam in the Severn River, and Tenterfield Creek forming different groups. 3D PCA plots showing the first three principal components are available in Supplementary Data 1–4.

## Discussion
Dispersal and gene flow are essential for the viability of natural populations, critical for ecological and evolutionary processes such as recolonisation, dispersal to suitable habitats, increased genetic diversity to avoid inbreeding depression and allow adaptation[26,29,30]. There is increasing concern about the impacts of dams on aquatic biota and ecological processes[15,17] given this is a critical global issue for rivers, with at least 2.8 million reservoirs larger than 0.1 ha[38]. Our analyses suggest that major dams pose barriers to platypus dispersal and gene flow given that genetic differentiation increased proportionally with time after

**Table 1 Summary statistics across the four river regions.**

| Region | River/Creek | Survey section (km) | Sample size | Proxy of abundance | Allelic richness | Ho | SE | He | SE | $F_{IS}$ | SE |
|---|---|---|---|---|---|---|---|---|---|---|---|
| Upper Murray Rivers | Ovens | 36 | 19 | 27 | 1.375 | 0.144 | 0.003 | 0.145 | 0.003 | 0.005 | 0.004 |
| | Mitta-Mitta above dam | 23 | 13 | 19 | 1.370 | 0.140 | 0.003 | 0.143 | 0.003 | 0.013 | 0.005 |
| | Mitta-Mitta below dam | 18 | 4 | 4 | 1.395 | 0.152 | 0.003 | 0.153 | 0.003 | −0.019 | 0.009 |
| Snowy Rivers | Snowy | 26 | 56 | 46 | 1.365 | 0.139 | 0.002 | 0.141 | 0.002 | 0.010 | 0.002 |
| | Thredbo | 33 | 19 | 37 | 1.365 | 0.141 | 0.003 | 0.141 | 0.003 | −0.003 | 0.004 |
| | Eucumbene above dam | 18 | 4 | 36 | 1.370 | 0.144 | 0.003 | 0.143 | 0.003 | −0.026 | 0.009 |
| | Eucumbene below dam | 20 | 20 | 50 | 1.346 | 0.137 | 0.003 | 0.136 | 0.003 | −0.001 | 0.004 |
| Central NSW Rivers | Wingecarribee* | 7 | 42 | ** | 4.113 | 0.703 | 0.060 | 0.731 | 0.044 | 0.053 | 0.047 |
| | Nepean above dam* | 0.5 | 11 | ** | 3.942 | 0.549 | 0.063 | 0.646 | 0.064 | 0.142 | 0.051 |
| | Nepean below dam* | 4 | 7 | ** | 4.706 | 0.589 | 0.095 | 0.608 | 0.059 | 0.096 | 0.107 |
| Border Rivers | Tenterfield | 96 | 39 | 207 | 1.353 | 0.135 | 0.003 | 0.138 | 0.003 | 0.015 | 0.003 |
| | Severn above dam | 50 | 23 | 115 | 1.335 | 0.133 | 0.003 | 0.133 | 0.003 | −0.003 | 0.004 |
| | Severn below dam | 60 | 17 | 83 | 1.333 | 0.130 | 0.003 | 0.131 | 0.003 | 0.005 | 0.005 |

The number of samples and a proxy of abundance calculated as (unique number of captures/number of sampling nights) × (length of the river surveyed) based on Hawke et al.[21]; Ho—observed heterozygosity; He—expected Hardy–Weinberg heterozygosity; $F_{IS}$—inbreeding coefficient.
Note that small sample sizes in Mitta-Mitta below the dam and Eucumbene above the dam (both 4 individuals) are likely to result in unreliable estimates of diversity.
SE standard error, NSW New South Wales.
*Microsatellite data.
**Comparable estimates are not available due to different survey techniques see Kolomyjec et al.[31,53,55].

**Table 2 Genetic differentiation ($F_{ST}$) between rivers in different connectivity scenarios.**

| Region | River 1 | River 2 | $F_{ST}$ | SE | Connectivity scenario |
|---|---|---|---|---|---|
| Border Rivers | Tenterfield | Severn above dam | 0.063 | 0.002 | Separated by a river system |
| | Tenterfield | Severn below dam | 0.075 | 0.002 | Separated by a river system |
| | Severn below dam | Severn above dam | 0.061 | 0.002 | Separated by dam for 47 years (*Circa* 1969)* |
| | Tenterfield above | Tenterfield below | 0.007 | 0.001 | No dam |
| Upper Murray Rivers | Ovens | Mitta-Mitta above dam | 0.052 | 0.002 | Contiguous river systems |
| | Ovens | Mitta-Mitta below dam | 0.035 | 0.003 | Contiguous river systems |
| | Mitta-Mitta above dam | Mitta-Mitta below dam | 0.024 | 0.003 | Separated by dam for 39 years (*Circa* 1979) |
| | Ovens above | Ovens below | 0.002 | 0.002 | No dam |
| Snowy Rivers | Snowy | Thredbo | 0.024 | 0.001 | Separated by dam for 50 years (*Circa* 1967) |
| | Snowy | Eucumbene above dam | 0.042 | 0.002 | Separated by dam for 59 years (*Circa* 1958) |
| | Snowy | Eucumbene below dam | 0.045 | 0.001 | Separated by dam for 50 years (*Circa* 1967) |
| | Thredbo | Eucumbene above dam | 0.040 | 0.003 | Separated by dam for 59 years (*Circa* 1958) |
| | Thredbo | Eucumbene below dam | 0.031 | 0.002 | Separated by lake for 50 years (*Circa* 1967) |
| | Eucumbene above dam | Eucumbene below dam | 0.053 | 0.003 | Separated by dam for 59 years (*Circa* 1958) |
| Central NSW Rivers | Wingecarribee** | Nepean above dam | 0.060 | 0.023 | Contiguous river systems |
| | Wingecarribee** | Nepean below dam | 0.062 | 0.013 | Contiguous river systems |
| | Nepean above dam** | Nepean below dam | 0.073 | 0.018 | Separated by dam for 74 years (*Circa* 1935) |
| | Wingecarribee above** | Wingecarribee below | 0.016 | 0.007 | No dam |

*SE* standard error.
*Pindari Dam. The height of the dam wall was doubled from 45 m to 85 m in 1995.
**Microsatellite data.

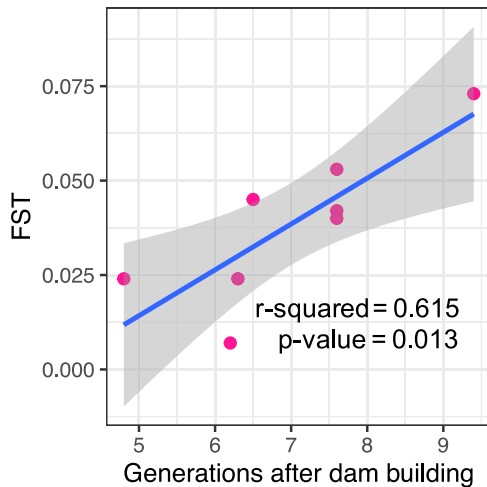

**Fig. 2 Genetic differentiation against dam age.** Relationship between genetic differentiation ($F_{ST}$) between platypus groups separated by major dams ($n = 8$ major dams) and the number of platypus generations (7.9 years[68]) since the building of the dam. Genetic differentiation increased at a rate of 0.011 per generation.

the building of a dam and was higher in dammed than undammed rivers.

In relation to whether major dams affect the connectivity between platypus groups, $F_{ST}$ values were higher when there was a dam, and some $F_{ST}$ values between groups separated by a dam were as high as $F_{ST}$ values between groups in different rivers (Table 2). In addition, we found a significant association between $F_{ST}$ and the number of platypus generations since dam construction (Fig. 2), suggesting that $F_{ST}$ increases at a rate of 0.011 by generation. Even though the Nepean dam, built in 1935, was analysed with a different type of molecular marker (microsatellites, not SNPs), recent research indicates that estimates of $F_{ST}$ using SNPs and microsatellites are comparable[39,40]. If anything, we would expect the microsatellites used in this system to have lower $F_{ST}$ due to the effect of their high within-group variation[37,41], but in fact the opposite trend was seen. We noticed

that $F_{ST}$ values in the Snowy Rivers were higher between groups separated by the Jindabyne Dam (Eucumbene below dam/Snowy; $F_{ST} = 0.045$) than between groups divided by the Jindabyne reservoir but not a dam (Eucumbene below dam/Thredbo; $F_{ST} = 0.031$). This observation suggests that some limited gene flow might have occurred across the Jindabyne reservoir.

Overall, our results are consistent with the notion that major dams and their associated waterbodies may be considerable barriers for platypuses. Despite platypuses being able to move substantial distances (e.g., male juveniles can move >40 km[42–44]), the effect of major dams on genetic differentiation was considerable. Such impacts can be directly related to the dam walls representing a barrier dissuading platypuses from attempting to bypass the wall through overland movements as well as indirectly by increasing predation risk by introduced predators such as foxes, cats, and dogs[24].

Major dams represent dispersal barriers for most freshwater species[45,46], requiring mitigation strategies to offset negative demographic impacts. For instance, human-mediated relocation of individuals between populations has been implemented successfully to limit the effects of population isolation and small population size[47]. A common rule of thumb in conservation suggests that one dispersing individual per generation would minimise the effects of population isolation[48]. Another strategy to improve connectivity between populations, despite some limitations and caveats, is the construction of dam bypass structures that increase dispersal of freshwater species, including fishways[49–51], although there are adverse consequences of connectivity, such as disease risks[52]. Such by-pass structures have not yet been considered for the platypus.

We have found that platypus connectivity between groups is adversely affected by major dams, and it is known that reduced connectivity can lead to the adverse long-term conservation outcomes described above[26–30]. Therefore there will be a need for the management of platypuses to consider ways such as those just described to minimise detrimental effects of river regulation on the platypus (and other species). Some of the long-term effects of major dams might be reduced by rare natural dispersal events between rivers[53], but our results indicate that this has not been enough to offset the divisive effect of the major dams, so more active management is required. Firstly, new dams within the

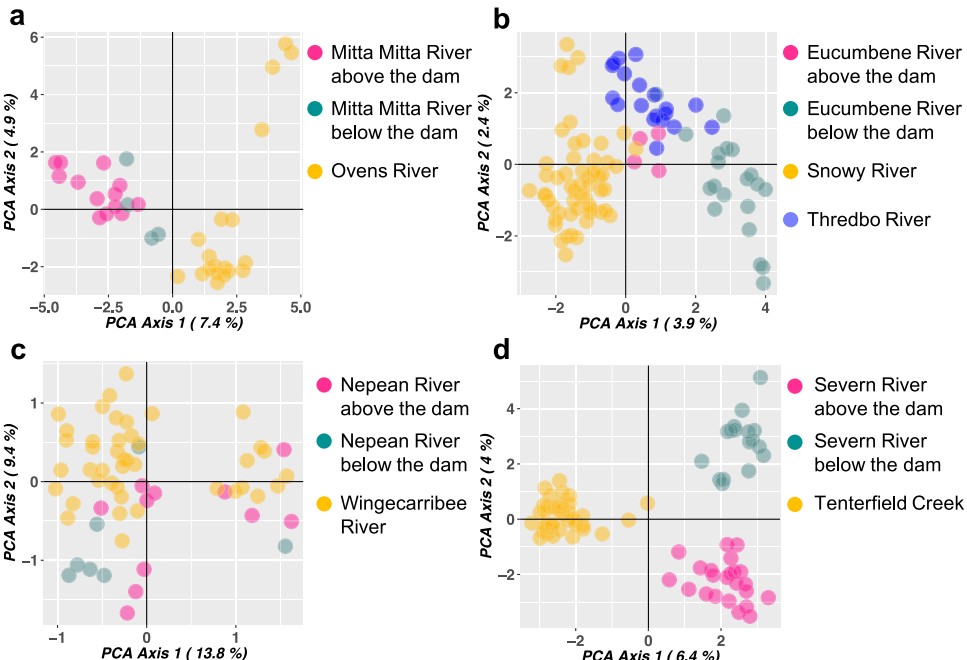

**Fig. 3 Principal coordinates analyses. a** Upper Murray Rivers: unregulated (no dam) Ovens and regulated (dam) Mitta-Mitta Rivers. **b** Snowy Rivers: regulated (dam) Snowy, unregulated (no dam) Thredbo and regulated (dam) Eucumbene Rivers. These rivers do not follow the paired experimental design due to geographic constraints. **c** Central NSW Rivers: regulated (dam) Nepean and unregulated (no dam) Wingecarribee Rivers. **d** Border Rivers: unregulated (no dam) Tenterfield Creek and regulated (dam) Severn River. Numbers between parentheses in the axis labels show the percentage of variation captured by each axis. Each point represents a platypus individual.

**Table 3 The four study systems and the major dams.**

| Region | River/Creek | Dam name | Year of completion | Dam height (m) | Dam volume (GL) |
|---|---|---|---|---|---|
| Upper Murray Rivers[c] | Ovens | – | – | – | – |
| | Mitta-Mitta | Dartmouth | 1979 | 180 | 3856 |
| Snowy Rivers[d] | Snowy | Jindabyne | 1967 | 72 | 688 |
| | Thredbo | – | – | – | – |
| | Eucumbene | Eucumbene | 1958 | 116 | 4798 |
| Central NSW Rivers[e] | Wingecarribee | – | – | – | – |
| | Nepean | Nepean | 1935 | 85 | 68 |
| Border Rivers[f] | Tenterfield | – | – | – | – |
| | Severn | Pindari* | 1969 | 85 | 312 |

See Fig. 1 for details of geography. The letters c, d, e and f refer to panels in Fig. 1.
GL gigalitres.
*Pindari Dam. The height of the dam wall was doubled from 45 m to 85 m in 1995.

platypus distribution need to be avoided, for example, by pumping from the river into an off-stream storage without the necessity for a dam on the river itself, as is done for metropolitan water supplies in both the Manning and Hastings Rivers, in New South Wales. Secondly, for existing major dams, it might be possible to devise platypus-specific versions of methods that have been used to ameliorate dam effects in other species, such as human-mediated relocation of individuals or by-pass structures that increase dispersal.

In this study, we compared regulated rivers, with major dams, to adjacent unregulated rivers with no major dams and identified that major dams were barriers to movement of platypuses within a river system, reflected in genetic variation. Major dams restricted dispersal and gene flow between groups and therefore increased the possibility of inbreeding depression, loss of adaptive genetic variation, failure to recolonise areas where local extinctions have occurred and failure to disperse to areas with more suitable conditions. Synergistic with reduced habitat quality, these

are all expected to lower the long-term viability of the platypus[11]. Our analyses reinforce the growing evidence on the negative impacts of major dams on platypus populations. These studies are relevant to inform the decision-making process of conservation managers and could be used in viability analysis and decision analysis[54] to develop strategies that ensure the long-term persistence of the unique platypus. This study adds to the growing evidence about the impacts of dams on aquatic biota and their viability.

## Methods

**Study areas and fieldwork**. Samples from platypuses were collected from nine different rivers (five regulated by major dams and four unregulated) across four regions in south-east Australia (see Fig. 1 and Table 3), also described in Hawke et al.[21] and Kolomyjec et al.[55,56]. River flows upstream of major dams were minimally regulated, contrasting with heavily regulated downstream flows. Throughout their range, the platypus comprises four major geographically defined genetic clusters: North Queensland, central Queensland, New South Wales and Tasmania[57]. The samples used in this study belong to the New South Wales cluster.

**Table 4 SNP filtering.**

| Filter | Variation between groups | Variation within groups |
|---|---|---|
| Reproducibility (RepAvg) | >100% | >100% |
| Retain only one SNP per read | Used | Used |
| Departure from Hardy-Weinberg proportions | <0.05 | <0.05 |
| Mapped to chromosome | Used | Used |
| BLAST alignment E-value | <1e−20 | <1e−20 |
| Missing data by site | >90% | >100% |
| Minor allele count (MAC) | >3 | Not used |
| Linkage disequilibrium ($r^2$) | <0.2 | Not used |
| Remove sites located within coding regions | Used | Not used |
| Remove sites located within sex chromosomes | Not used | Used |
| Total SNPs after filtering | 2641 | 4551 |

Filters and their thresholds used for SNPs to remove genomic sites for the analyses based on variation between groups and variation within groups.

Platypuses were captured across 81 sites (Fig. 1). In this study, we used two different molecular markers: single nucleotide polymorphism (SNPs) for all samples except Central NSW, and microsatellites for Central NSW[55,56]. Sampling for microsatellites in Central NSW is described in Kolomyjec et al.[55,56]. For SNPs at all other sites, we aimed to cover a minimum of 40 km of each unregulated river and 20 km of river above and below major dams on regulated rivers. The procedure of trapping and sampling platypuses, including details of anaesthesia, used in this study have been described elsewhere[21,58]. Briefly, platypuses were captured using fyke nets or unweighted mesh (gill) nets and implanted with a Passive Integrated Transponder (PIT) tag (Trovan) to identify recaptured individuals. Platypuses were then weighed, measured, sexed, aged, and blood collected (~2 ml) and stored in Qiagen RNAprotect® animal blood tubes (Qiagen, Hilden, Germany). For the SNP sampling, our proxy of abundance for each river was the following metric: unique number of captures/number of sampling nights x length of the river surveyed (see Hawke et al.[21]).

**Laboratory work.** For SNPs (single nucleotide polymorphisms), genomic DNA was extracted from whole blood using a Qiagen DNeasy Blood and Tissue kit (Qiagen, Hilden, Germany). DNA quality and concentration were visualised using agarose gel electrophoresis and quantified fluorimetrically with a Qubit 2.0 (Thermo Fisher Scientific). Samples were genotyped using DArTseq[TM] (DArT Pty Ltd, Canberra, ACT, Australia). DArT's procedure uses a combination of genome complexity reduction methods using restriction enzymes, implicit fragment size selection and next-generation sequencing to produce thousands of SNPs randomly distributed throughout the genome[59]. Read sequences were processed using proprietary DArT analytical pipelines[59] and mapped to the representative platypus genome (mOrnAna1.p.v1, GenBank assembly accession: GCA_004115215.2; total sequence length of 1.8 Gbps, 305 scaffolds with an N50 of 83 Mbp). Refer to Georges et al.[60] for details of DArT sequencing, genotyping and filtering processes. DArT's genotyping has various advantages such as limiting the potential for ascertainment bias[61], providing metadata for each locus with various quality and BLAST alignment measures, including the proportion of replicates for which the marker score is consistent (RepAvg) and the average of the polymorphism information content of the reference and SNP allele (AvgPIC).

For microsatellites, genomic DNA was extracted from toe-web biopsies (2 × 2 mm specimens stored in 70% ethanol) using a proteinase K/salt precipitation method[62]. Twelve published microsatellite sites were amplified and scored according to standard techniques[55,56].

**SNP filtering.** The criterion for SNP filtering used to analyse variation between groups (e.g., $F_{ST}$) can bias estimates of variation within groups (e.g., heterozygosity[63]). Therefore, we used different filtering settings for each type of analysis (Table 4). Detailed description of the filtering processes can be found in the Supplementary Information document.

For SNPs, a total of 295 platypuses were captured and blood sampled across four river regions in southeast Australia (Supplementary Table 2). DNA extraction and DArT[TM] sequencing were successful in 218 blood samples (Supplementary Table 2). Two samples, each collected in a different river (V30 in Ovens and V32 in Mitta-Mitta), showed contrasting genetic patterns relative to samples collected in the same river (Supplementary Figure 1). Relatedness analyses performed in the R package *related*[64] revealed these two samples had closer relatives in the opposite river (Supplementary Table 1). In addition, the locations of these two samples were separated by 46 km, steep mountainous terrain, and a river system. Under these conditions, we considered that dispersal events were unlikely and concluded that samples were mislabelled and therefore assigned them to the presumed correct river and site. Relatedness analyses also identified two pairs of samples in which each pair was collected from the same individual (i.e., recaptures; samples T3-T5 and T28-T42; Supplementary Table 1). Consequently, we removed one sample from each pair. In the unlikely event that these were pairs of identical twins, it would still be appropriate to remove one of each pair.

For SNPs, sequencing provider DArT[TM] (Canberra) successfully genotyped 17,631 single nucleotide polymorphism (SNP) sites. After stringent filtering, our dataset for analysing genetic variation between groups comprised 2641 SNPs genotyped in 214 platypus samples (108 females, 106 males). After filtering, our SNP dataset for analysing genetic variation within groups comprised 4551 SNPs genotyped in 214 platypus samples (108 females, 106 males).

**Data analyses**

*Genetic variation within groups.* To measure genetic variation within rivers, we calculated observed heterozygosity (Ho), expected heterozygosity (He) and allelic richness using the R package *Hierfstat*[65]. After identifying that the data did not conform to a normal distribution, using a Shapiro–Wilk test of normality (R function *shapiro.test*), we tested whether He was significantly different between groups using a non-parametric Mann–Whitney U test (R function *wilcox.test* with option paired = FALSE). In addition, we calculated the inbreeding coefficient ($F_{IS}$) of each river group using *Hierfstat*.

*Investigating whether major dams affect connectivity between platypus groups.* We used multiple approaches to investigate whether major dams affect gene flow between platypus groups. Firstly, to test whether groups separated by major dams are more genetically different than otherwise, we divided the sampling sites of each pair of rivers into comparable upstream and downstream groups. For regulated rivers (Nepean, Severn and Mitta-Mitta), the dam, ignoring the reservoir, was used as reference point for the division. For unregulated rivers (Wingecarribee, Tenterfield and Ovens), the division point was chosen at a comparable position to the dam in the paired regulated river. We then calculated the genetic differentiation using $F_{ST}$ following Nei's method[66] between the two groups within each river. We tested the significance of the difference of $F_{ST}$ values between dammed and unregulated rivers using a Mann–Whitney U test (R function *wilcox.test* with option paired = FALSE). In addition, we used Mutual Information[41] and Jost's D[67] two measures that assess between-group differentiation independently of within-group variation.

Secondly, to test whether the number of platypus generations since the building of the dams can predict the genetic differentiation of SNPs and microsatellites between groups ($F_{ST}$), we used univariate linear regression models (R function *lm*). We considered one platypus generation to be 7.9 years based on Pacifici et al.[68], who used information on age at first reproduction and reproductive life span to estimate generation length in platypus.

Thirdly, to visualise the spatial distribution of genetic variation of the sampled individuals, we performed principal component analysis (PCA) using the R package *dartR*[69] using our two datasets of SNP's and microsatellites. PCA is a statistical method that summarises the variance in the data and projects the top principal components onto a series of orthogonal axes[70]. We chose to use PCA because it has an exact mathematical relationship to the biological coalescent, or genealogy[70], and provides two-dimensional and three-dimensional displays, which are not available in other methods such as STRUCTURE[71].

*Statistics and reproducibility.* Sample sizes and statistical parameters used in each analysis are indicated in the relevant 'Methods' and 'Results' sections, as well as in tables when applicable. All statistical analyses were performed in R (v4.0.5)[72].

**Reporting summary.** Further information on research design is available in the Nature Research Reporting Summary linked to this article.

## Data availability

The datasets used for this research work are stored in GitHub: https://github.com/mijangos81/Platypus and have been archived within the Zenodo repository: https://doi.org/10.5281/zenodo.703977[73].

## Code availability

The R scripts used for this research work are stored in GitHub: https://github.com/mijangos81/Platypus and have been archived within the Zenodo repository: https://doi.org/10.5281/zenodo.7039778[73].

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

## Acknowledgements

J.L.M. was funded by a University College Postgraduate Research Scholarship from UNSW Canberra. This study was funded by ARC Linkage LP150100093, the Taronga Conservation Society, and the Australian Government's Environmental Water Holder. Trapping and handling of platypuses for the SNP samples were carried out in accordance with guidelines and approved by the NSW Office of Environment and Heritage (SL101655), NSW Department of Primary Industries (P15/0096-1.0 & OUT15/26392), and UNSW's Animal Care and Ethics Committee (16/14A). Trapping and handling of platypuses for the microsatellite samples from the Nepean and Wingecarribee Rivers was approved under the [then] NSW Department of Environment and Climate Change Scientific Research License number S10478, NSW Department of Primary Industries (DPI) Scientific Research Permit number F84.1245 and NSW DPI Animal Research Authority - Trim File No. 01/1091. Consent to enter Special Metropolitan [catchment] Areas was provided by the [then] Sydney Catchment Authority (Reference No. 3008/07934; 2009). We are grateful to Dr. Clare Holleley, Dr. Melody Serena, Dr. Simon Watt, Jia Zhou and Alex Sentinella for their valuable comments that importantly improved the manuscript.

## Author contributions

W.B.S., G.B., R.T.K., J.G. and others conceived the project and acquired the research funds; H.S. and W.B.S. supervised; J.L.M., J.D., K.N.D. and J.G. performed DNA extraction; G.B., T.H. and T.G. carried out fieldwork and collected samples; S.H.K. performed the microsatellite analyses; J.L.M. analysed the data; J.L.M. wrote the manuscript with support from W.B.S. T.G. and S.H.K. first pondered on the possible effect major dams may be having on the genetics of the platypus, carried out the initial research and encouraged others to further investigate this aspect of platypus conservation. All authors discussed the results and contributed to the final manuscript.

## Competing interests

The authors declare no competing interests.
