## [Peer Review File · Communications Biology]

Reviewers' comments:

Reviewer #1 (Remarks to the Author):

The authors of this MS examined the genetic differentiation between platypus groups below and above major dams using a combination of SNPs and microsatellites. The genetic differentiation, measured by F_{ST} , across dams was found to be 4- to 20- fold higher than along similar stretches of adjacent undammed rivers. F_{ST} between populations was also found to be significantly correlated with the year in which the dam was built. It is concluded that dams pose a serious barrier to the gene flow of platypus populations. The study has important implications for the general conservation management of platypus populations. Overall, the MS is well written and is worth publishing, in my opinion, in *Communications Biology*.

I have just a few minor comments for the authors to consider in revision.

1. Lines 135-138, "Genetic differentiation can increase due to dispersal barriers within one to 15 generations during simulations (Landguth et al., 2010), but is unlikely to arise if population size is large (> 50 individuals) or if the species lifespan is long (> 22 years; Hoffman et al., 2017)." Genetic differentiation due to drift is determined by the number of generations and the population size. The species lifespan is irrelevant for a given number of generations.
2. Too much is spent on SNP filtering, from line 197 to 278. I know how SNP is filtered before an analysis is important, but that is not the focus of a study. So I suggest either shortening this section or moving it to an online supplementary appendix.
3. About data analyses, I wonder why the authors choose to use PCA (described in line 309-317) rather than a population genetics model-based method such as STRUCTURE. With a few thousands of loci (SNPs), it is not difficult to conduct STRUCTURE analysis, which gives results more readily interpretable biologically than PCA.
4. About genetic variation within groups (from line 320 to 328), I wonder how the authors deal with unequal sample sizes which are shown in Table 3. Without correction for unequal sample sizes, then small samples would tend to have a smaller genetic variation measure either as H_e or allelic richness.
5. Line 403-404, "If anything, we would expect the microsatellites used in this system to have lower F_{ST} due to the effect of their high within-population variation". I do not agree that we should always expect microsatellites to have lower F_{ST} (than SNPs) just because of their high within-population variation. Indeed, microsatellites have usually more alleles and higher within population variation than SNPs, because of their higher mutational rate. However, this does not necessarily lead to an underestimation of F_{ST} . The estimated F_{ST} values from markers depend on the relative effects of drifts, migrations, and mutations (if other forces such as selection can be ignored). F_{ST} is still estimated unbiasedly by microsatellites when either drift or migration is strong relative to mutation rate. F_{ST} is underestimated when large populations are in isolation or have very small migration rates for a long time such that mutations kick in.

Reviewer #2 (Remarks to the Author):

This MS reports the first comprehensive analysis of the impact of major dams on gene flow and dispersal in the platypus. The novel finding of disruption to gene flow by major dams is highly significant and will be of broad interest conservation biologists, river ecologists and wildlife managers. This finding will challenge current thinking re the impact of dams on similar semi-aquatic species. The sampling, experimental design and analysis of the data is rigorous and appropriate. The one potential weakness is that the data are not based on a consistent set of genetic markers which is unfortunate and does introduce an element of uncertainty which the authors have tried to address. Nevertheless the microsat data from a single river system is consistent with the more robust SNP data from multiple river systems.

The MS does not spend much time establishing or discussing a global context. Or discussing the worldwide data available on impact of dams on genetic connectivity of aquatic and semi-aquatic

species. This would strengthen its relevance and impact

Some improvements in wording and presentation could be made.

1. Page 1, In 24 and elsewhere. 'groups' seems to be mostly used for individuals sampled at a site or cluster of sites throughout instead of the more familiar 'populations'. I understand the distinction and lack of assumptions that the authors are trying to make but some greater clarity and explanation would aid comprehension. And this is then further confused by the a priori grouping of samples into 'populations' to test for HWE during filtering.

2. Page 2 In 40. 'Species-poor' may be clearer

3. Pg2, In 40/41 'basal branch of mammals' would be simpler

4. pg2, In 46-47 I understand the point the authors are trying to make the wording here needs some work. There have been now extinct monotremes during this period and all monotremes and indeed all mammals have the same length of evolution.

5. pg2 In 52 habitat loss is just or more important than habitat degradation

6. Page 3 In 62 'and' is missing

7. Page 3 In64 'habitat loss and degradation'

8. pg3 In 68-70 meaning unclear as sentence too long and complicated. at this stage of the MS dams are potential threats

9. pg3 In 72. apostrophe missing from platypus or reword to avoid need to establish possession

10. pg 5 In 126 'dam is also unclear' would be better

11. pg7 In175 'Genomic' should be 'genomic'

12. pg8 In 193 ', genomic DNA'

13. fig 3. For comparison, PCA plots need to show % variation explained on both X and Y axes.

14. pg20 In417-419. This one point of comparison with the published literature is rather perfunctory and confusing. Do you mean the difference is that blackfish have a large N_e ? cf platypus?

Note that when the option of track changes in Word is activated, line numbers are not continuous between pages.

Reviewer #1:

The authors of this MS examined the genetic differentiation between platypus groups below and above major dams using a combination of SNPs and microsatellites. The genetic differentiation, measured by F_{ST} , across dams was found to be 4- to 20- fold higher than along similar stretches of adjacent undammed rivers. F_{ST} between populations was also found to be significantly correlated with the year in which the dam was built. It is concluded that dams pose a serious barrier to the gene flow of platypus populations. The study has important implications for the general conservation management of platypus populations. Overall, the MS is well written and is worth publishing, in my opinion, in *Communications Biology*.

I have just a few minor comments for the authors to consider in revision.

1. Lines 135-138, "Genetic differentiation can increase due to dispersal barriers within one to 15 generations during simulations (Landguth et al., 2010), but is unlikely to arise if population size is large (> 50 individuals) or if the species lifespan is long (> 22 years; Hoffman et al., 2017)." Genetic differentiation due to drift is determined by the number of generations and the population size. The species lifespan is irrelevant for a given number of generations.

We agree with the reviewer comment. We removed the sentence (revised ms lines 190): "or if the species lifespan is long (> 22 years)".

2. Too much is spent on SNP filtering, from line 197 to 278. I know how SNP is filtered before an analysis is important, but that is not the focus of a study. So I suggest either shortening this section or moving it to an online supplementary appendix.

As suggested by the reviewer, the description of the filtering process was moved to the Supplementary Information document. We added the following text (revised ms lines 254-255):

"Detailed description of the filtering processes can be found in the Supplementary Information document."

3. About data analyses, I wonder why the authors choose to use PCA (described in line 309-317) rather than a population genetics model-based method such as STRUCTURE. With a few thousands of loci (SNPs), it is not difficult to conduct STRUCTURE analysis, which gives results more readily interpretable biologically than PCA.

We chose to use PCA because it has an exact mathematical relationship to the biological coalescent, or genealogy (McVean, 2009), and provides a two-dimensional display, which is not available in other methods such as STRUCTURE (Pritchard *et al.*, 2000). The two-dimensional display is important for the more complex situation in the Snowy Rivers, and we wish to avoid using multiple methods to address the same

question, so we did not do STRUCTURE. We added the following text to highlight this issue (revised ms lines 415-419):

“We chose to use PCA because it has an exact mathematical relationship to the biological coalescent, or genealogy (McVean, 2009), and provides a two-dimensional display, which is not available in other methods such as STRUCTURE (Pritchard et al., 2000).”

4. About genetic variation within groups (from line 320 to 328), I wonder how the authors deal with unequal sample sizes which are shown in Table 3. Without correction for unequal sample sizes, then small samples would tend to have a smaller genetic variation measure either as H_e or allelic richness.

Smaller sample sizes result in under-representation of some rare alleles. Heterozygosity (H_e) will be largely unaffected by this under-representation, because it chiefly represents variation at common alleles (Sherwin *et al.*, 2017). Moreover, the focus of the article is not within-location variation such as H_e or allelic richness, but between-location variation.

5. Line 403-404, “If anything, we would expect the microsatellites used in this system to have lower F_{ST} due to the effect of their high within-population variation”. I do not agree that we should always expect microsatellites to have lower F_{ST} (than SNPs) just because of their high within-population variation. Indeed, microsatellites have usually more alleles and higher within population variation than SNPs, because of their higher mutational rate. However, this does not necessarily lead to an underestimation of F_{ST} . The estimated F_{ST} values from markers depend on the relative effects of drifts, migrations, and mutations (if other forces such as selection can be ignored). F_{ST} is still estimated unbiasedly by microsatellites when either drift or migration is strong relative to mutation rate. F_{ST} is underestimated when large populations are in isolation or have very small migration rates for a long time such that mutations kick in.

The reviewer focuses on the processes that affect the variation, but these processes are powerless to overcome the limitation of F_{ST} that although it is often used as a between-location measure (as we have done) its value also depends strongly on within-location variation. For example, if a SNP has no shared alleles between two locations, and only two alleles (one at each location), then $F_{ST} = 1$. But if a microsatellite has no shared alleles between two locations, but three equally-frequent alleles at each location, then $F_{ST} = 0.2$, not the expected unity.

See Sherwin et al. (2017) and many references therein dating back to the 1970s, to confirm this.

Reviewer #2

This MS reports the first comprehensive analysis of the impact of major dams on gene flow and dispersal in the platypus. The novel finding of disruption to gene flow by major dams is highly significant and will be of broad interest conservation biologists, river ecologists and

wildlife managers. this finding will challenge current thinking re the impact of dams on similar semi-aquatic species. The sampling, experimental design and analysis of the data is rigorous and appropriate. The one potential weakness is that the data are not based on a consistent set of genetic markers which is unfortunate and does introduce an element of uncertainty which the authors have tried to address. Nevertheless the microsat data from a single river system is consistent with the more robust SNP data from multiple river systems.

Yes, it was unfortunate that one dataset had a different basis, but as explained in the MS at line 403-404 submitted (revised ms lines 515-518), the possible bias in direction of results was not observed - in fact the difference between the two types of data was in the opposite direction.

The MS does not spend much time establishing or discussing a global context. Or discussing the worldwide data available on impact of dams on genetic connectivity of aquatic and semi-aquatic species. This would strengthen its relevance and impact.

Following reviewer's suggestion, we have added the following text in the introduction and discussion sections:

Introduction (revised ms lines 80-88):

“Threats to freshwater ecosystems are commonly synergistic and are intensified by the construction of major dams that can have immediate and long-term impacts (Reid et al., 2019). Nearly half of the world's river discharge is impacted by flow regulation and fragmentation (Grill et al., 2015). Dams pose a significant threat to global freshwater biodiversity (Winemiller et al., 2016). Large dams form major barriers for aquatic organisms, limiting critical ecological processes, such as fish migration (Dugan et al., 2010). Water impoundments behind major dams form wind-exposed, deep, and standing (lentic) ecosystems which can offer little resources for flow dependant species (Timpe & Kaplan, 2017)”.

Discussion (revised ms lines 501-504):

“There is increasing concern about the impacts of dams on aquatic biota and ecological processes (Winemiller et al., 2016; Reid et al., 2019) given this is a significant global issue around the world for rivers, with at least 2.8 million reservoirs larger than 0.1 ha (Lehner et al., 2011)”.

Some improvements in wording and presentation could be made.

1. Page 1, ln 24 and elsewhere. 'groups' seems to be mostly used for individuals sampled at a site or cluster of sites throughout instead of the more familiar 'populations'. I understand the distinction and lack of assumptions that the authors are trying to make but some greater clarity and explanation would aid comprehension. And this is then further confused by the a priori grouping of samples into 'populations' to test for HWE during filtering.

"Population" has a specific meaning of a distinct random-mating unit, which we cannot confirm in all cases. We used this term for generic meanings, such as "population viability analysis" etc. However, because of the inability to confirm the complete population boundaries according to the definition above, we aimed to use the term "groups" whenever talking about samples that we have taken from geographically close locations, such as above a dam. Nevertheless, as pointed out by the reviewer, we occasionally called these "populations" by mistake - our apologies. We have now amended the manuscript so that whenever we are talking about "samples that we have taken from geographically close locations, such as above a dam", we use the term "groups".

2. Page 2 In 40. 'Species-poor' may be clearer

As suggested, we changed (revised ms lines 43) "species-scarce" to "species-poor".

3. Pg2, In 40/41 'basal branch of mammals' would be simpler

As suggested, we changed (revised ms lines 44) "basal branch of the mammalian group" to "basal branch of mammals".

4. pg2, In 46-47 I understand the point the authors are trying to make the wording here needs some work. There have been now extinct monotremes during this period and all monotremes and indeed all mammals have the same length of evolution.

We agree with the reviewer, we changed the text to make clearer our point. The text now reads (revised ms lines 48-50):

"The uniqueness and rarity of platypus features (sensu Pavoine et al., 2005) and its evolutionary distinctiveness (Isaac et al., 2007) make it arguably one of the most irreplaceable mammals existing today."

5. pg2 In 52 habitat loss is just or more important than habitat degradation

We agree with the reviewer. We have added the following text (revised ms lines 58):

"loss and modification of habitats".

6. Page 3 In 62 'and' is missing

We added the word (revised ms lines 53) "and" as suggested.

7. Page 3 In64 'habitat loss and degradation'

We changed the text as suggested. The text now reads (revised ms lines 58):

"loss and modification of habitats"

8. pg3 ln 68-70 meaning unclear as sentence too long and complicated. at this stage of the MS dams are potential threats add potential

We shortened the sentence and made it clearer and added the word “can”. Now the text reads (revised ms lines 80-81):

“Threats to freshwater ecosystems are commonly synergistic and are intensified by the construction of major dams that can have immediate and long-term impacts (Reid et al., 2019).”

9. pg3 ln 72. apostrophe missing from platypus or reword to avoid need to establish possession

The apostrophe was added as suggested (revised ms lines 91):

“Major dams are widespread across much of the platypus’ distribution”

10. pg 5 ln 126 'dam is also unclear' would be better

We added the suggested text (revised ms lines 172):

“Their ability to swim across the large deep-water impoundments above the dam is also unclear”.

11. pg7 ln175 'Genomic' should be 'genomic'

The word was changed (revised ms lines 227) from “Genomic” to “genomic”.

12. pg8 ln 193 ', genomic DNA'

The word “genomic” was added (revised ms lines 246).

13. fig 3. For comparison, PCA plots need to show % variation explained on both X and Y axes.

As suggested, we added the percentage of variation to each PCA plot (revised ms lines 483).

14. pg20 In 417-419. This one point of comparison with the published literature is rather perfunctory and confusing. Do you mean the difference is that blackfish have a large N_e ? cf platypus?

We agree with the reviewer. We removed this comparison.

References

- McVean, G. (2009). A genealogical interpretation of principal components analysis. *PLoS Genet*, 5(10), e1000686. doi:10.1371/journal.pgen.1001553
- Pritchard, J. K., Stephens, M., & Donnelly, P. (2000). Inference of population structure using multilocus genotype data. *Genetics*, 155(2), 945-959.
- Sherwin, W. B., Chao, A., Jost, L., & Smouse, P. E. (2017). Information theory broadens the spectrum of molecular ecology and evolution. *Trends in Ecology & Evolution*, 32(12), 948-963.

REVIEWERS' COMMENTS:

Reviewer #1 (Remarks to the Author):

I have read the revised version of the MS and the author's responses to referee's comments. Although a bit disappointed that some of my comments were dismissed without adequate justification (e.g. my suggestion of a STRUCTURE analysis against PCA, especially considering that the 1st two components explain less than 10% of the total variation!), I do not insist on addressing these comments. Overall this is a good paper worth publishing. I do not have further major comments on the revision.

Reviewer #2 (Remarks to the Author):

The revised MS is much improved and the authors have, in my opinion, successfully addressed the comments of both reviewers.